# Carbon Instrumentation in Patients with Metastatic Spinal Cord Compression

**DOI:** 10.3390/cancers16040736

**Published:** 2024-02-09

**Authors:** Søren Schmidt Morgen, Emma Benedikte Alfthan Madsen, Anders Skive Weiland, Benny Dahl, Martin Gehrchen

**Affiliations:** Spine Unit, Department of Orthopaedic Surgery, Rigshospitalet, University of Copenhagen, DK-2100 Copenhagen, Denmarkanders.weiland.skive@regionh.dk (A.S.W.); bennydahl@gmail.com (B.D.); pedicle@mac.com (M.G.)

**Keywords:** spine surgery, metastatic spinal cord compression, spinal metastasis, cancer treatment

## Abstract

**Simple Summary:**

Patients with metastatic spinal cord compression (MSCC) are usually treated with stabilizing surgery with implants and subsequent radiotherapy. Recently, spinal implants consisting of carbon (CI) instead of titanium (TI) has been introduced. This is expected to decrease the deflection of radiation and thereby improve diagnostic imaging. However, we do not know whether it is equally safe and effective to use CI instead of TI in MSCC patients. The aim of this study was to examine the safety and effectiveness of CI in patients with MSCC. We compared 80 patients stabilized with CI versus 83 with TI. The peri-operative blood loss in the CI-group was significantly lower than in the TI-group. There were no significant differences between the groups with regard to mean survival, mean BMI, mean ASA-score, or the number of patients with revisions. Surgical treatment with CI for MSCC is safe and an equally sufficient treatment when compared to TI.

**Abstract:**

Recently carbon spinal implants have been introduced in the treatment of patients with metastatic spinal cord compression (MSCC). This is expected to decrease the deflection of radiation and improve diagnostic imaging and radiotherapy when compared to titanium implants. The aim of this study was to determine the safety and effectiveness of spinal carbon instrumentation (CI) in patients with MSCC in a large cohort study. A total of 163 patients received instrumentation between 1 January 2017 and 31 December 2021. A total of 80 were stabilized with CI and 83 with TI. The outcome measures were surgical revision, postsurgical survival, peri-operative bleeding, and surgery time. The peri-operative blood loss in the CI-group was significantly lower than that in the TI-group: 450mL vs. 630mL, (*p* = 0.02). There were no significant differences between the groups in mean survival (CI 9.9) vs. (TI 12.9) months (*p* = 0.39), or the number of patients needing a revision (CI 6) vs. (TI 10), (*p* = 0.39). The median duration of surgery was 121 min, (*p* = 0.99) with no significant difference between the two groups. Surgical treatment with CI for MSCC is safe and an equally sufficient treatment when compared to TI.

## 1. Introduction

In recent decades there has been an increasing number of cancer patients who develop spinal metastases. Most likely, this development is caused by improvements in cancer treatment resulting in an increased life expectancy [1]. Metastatic spinal cord compression (MSCC) is defined as a compression of the spinal cord or cauda equina due to metastatic tissue. This may result in pain, instability, and neurological disability, such as paresthesia, paraplegia, and the loss of sphincter function [2].

It has been widely accepted that in some patients, the optimal treatment of MSCC is a combination of surgical stabilization with decompression and subsequent radiotherapy [3,4]. Titanium implants, which have been widely used due to their reliability and corrosion resistance, have a density significantly different from the density of the human body because of the larger atomic amount of titanium. These titanium-based implants are, consequently, known for creating a scattering effect on, e.g., MRI, resulting in artifacts that interfere with post-surgery imagining, and potentially also the radiation therapy. The presence of these artifacts can result in imprecise dose calculation and an insufficient quality of follow-up imagining [5]. It could be speculated that this could lead to the postponed discovery of tumor regrowth. 

Recently, spinal implants consisting of carbon material have been introduced as an alternative to titanium implants. The CarboClear system consists of pedicle screws, rods, and locking elements made entirely out of carbon-fiber-reinforced polyetheretherketone (CFR-PEEK). One of the qualities of the CFR-PEEK system is its radiographic translucency caused by the lower atomic number of the carbon implants, resulting in radiation properties more like those of human tissue. This is expected to decrease the deflection of radiation to the affected area of the spine and, consequently, improve not just the diagnostic imaging but also post-surgery radiotherapy, potentially leading to a superior cancer treatment [6,7]. 

It has been demonstrated in several studies that the mechanical properties of the CFR-PEEK system are comparable to those of titanium-based systems. This was determined via an in vitro mechanical evaluation examining fatigue resistance, torsional stiffness, axial pull strength, and the bending load of the CFR-PEEK system [8]. Moreover, this study showed superior fatigue qualities in the CFR-PEEK system in comparison to those of the titanium systems. Additionally, a biomechanical revision of screw loosing comparing CFR-PEEK and titanium showed that the carbon screws resisted an equivalent amount of load cycles compared to those of titanium [6]. In patients with MSCC, only a few studies have investigated the safety and effectiveness of using carbon instrumentation compared to titanium instrumentation [5,9,10,11,12]. Previous studies are limited by the lack of a control group and a small sample size.

Here, we designed a consecutive, single-center, cohort study to examine the safety and effectiveness of spinal carbon instrumentation compared to titanium instrumentation in patients with MSCC. 

## 2. Materials and Methods

All patients included in this cohort study were surgically treated at the Spine Unit, Department of Orthopedic Surgery, Rigshospitalet, Denmark. From the 1 January 2017 to the 31 December 2021, a total of 473 patients underwent surgical treatment for MSCC and 163 were included in the study. 

The inclusion criteria were as follows: patient age >18 years and the presence of metastatic spine tumors resulting in MSCC. Additionally, only patients where survival status could be obtained at the two-year follow-up period and who had received surgical treatment were included. The CI system we use is not designed for cervical instrumentation. CI rods are rigid and come with various prebend curvatures. They cannot always be precisely adapted to the body’s anatomy as it can be done with TI. This can be a particular challenge, especially in instrumentation at the pelvis or high thoracic area where there is increased kyphosis. For this reason, only patients who received instrumentation between T5 and L5 were included. The two surgical procedures are illustrated in Figure 1a,b.

Out of the original 473 MSCC patients, 310 were excluded, and the final number of patients included in this cohort study was 163. A flowchart illustration of the exclusion process is shown in Figure 2.

Patient information regarding age, gender, BMI and ASA-score as shown in Table 1. Primary oncologic diagnosis was collected at baseline and adjusted in concordance with results of the pathological peri-operative tissue samples made during surgery as shown in Table 2. The outcome measures were surgical revision, postsurgical survival (days), peri-operative bleeding (mL), and surgery time (min).

A total of 80 of the elected patients were stabilized using CFR-PEEK implants (CI-group), while 83 patients were stabilized with titanium implants (TI-group). All patients were treated with posterior pedicle screw instrumentation and decompression at the metastatic level. The standard treatment was instrumentation 2 levels above and 2 levels below the metastatic level. The spinal cord was decompressed at the metastatic level with a wide laminectomy. In all cases, tissue samples were sent for analysis at the pathological department and primary cancer diagnoses were registered if not known beforehand. The choice of the surgical treatment was at the discretion of the surgeon. Diagnosis, primary tumor, and instrumentation are shown in Table 2. 

Mann–Whitney and log rank tests were used to compare the groups, and a *p*-value < 0.05 was considered statistically significant.

## 3. Results

Of the 163 patients included in this cohort study, 46.5% (*n* = 76) were women and 53.4% (*n* = 87) were men (*p* = 0.317). In the patient group that had been stabilized using CFR-PEEK implants (CI-group), 50% were women (*n* = 40) and 50% were men (*n* = 40), while the group that had been stabilized using titanium implants (TI-group) consisted of 43% (*n* = 36) women and 57% (*n* = 47) men. There were no significant differences between the groups with regard to average age, mean BMI, mean ASA-score, or gender, as illustrated in Table 1.

Metastatic pulmonary cancer was the dominating primary cancer, followed by metastatic breast cancer, and metastatic renal cancer. The distribution of primary oncologic diagnosis in the two patient groups is illustrated in Table 2.

The peri-operative blood loss in the CI-group was significantly lower than in the TI-group; the mean blood loss was 450 mL for the CI-group (range 100 mL–1800 mL) vs. 630 mL (range 150 mL–4100 mL) for the TI-group (*p* = 0.024). The overall median duration of surgery was 121 min. The surgery time ranged from 73 to 202 min for the CI-group, whereas the range in the TI-group was 67–329 min (*p* = 0.990).

There were six surgical revisions in the CI-group and 10 in the TI-group (*p* = 0.386). An overview of the causes of surgical revision and the distribution between the two groups is illustrated in Table 3.

In the CI-group, the mean two-year survival post surgery was 9.9 months, with the survival time ranging from 0.4 to 40.1 months. Out of the original 80 patients in the CI-group, 24 were still alive at the two-year follow-up mark. The mean two-year survival in the TI-group was 12.9 months, with a range of 0.4–38.7 months, and 24 patients out the 83 patients in the group were still alive at the two-year follow-up mark. The difference in survival time seems clinically relevant, but did not reach statistical significance (*p* = 0.388).

## 4. Discussion

The results of the present study indicate that the use of CFR-PEEK spinal implants in patients with MSCC is equally safe and effective compared to titanium-based implants with regard to complications and survival. We found that the peri-operative blood loss in the CI-group was significantly lower than in the TI-group. It could be speculated that this was caused by the fact that the use of CI vs. TI was at the surgeon’s discretion, and that the more experienced surgeons tended to choose the CI implant. There were more patients in the TI group who had two levels of metastases. These patients underwent instrumentation at six levels instead of five levels. This could explain some of the difference in blood loss between the two groups, since longer constructs and longer wounds potentially lead to a higher blood loss. There were no significant differences between the groups in mean survival time or the number of patients needing surgical revision, which speaks against surgical technique as the explanation for the reduced blood loss in the CI-group.

The aim of the present study was to examine the safety and effectiveness of carbon-based instrumentation for MSCC patients in a larger sample size than previously been published in the literature.

In a new systematic review by Khan et al., data were collected from all articles describing the treatment outcome on MSCC patients who underwent surgical stabilization with CFR-PEEK implants. In that study, they identified a total of 206 patients treated with CFR-PEEK implants, and those were compared with 47 patients treated with titanium. Khan et al. concluded that there is a need for direct comparable studies with larger sample sizes of patients, which was the aim of our study [11].

Radiation-sensitive metastases are treated postoperatively with radiation therapy. The goal is to target the tumor tissue with radiation, and, at the same time, keep the doses as low as possible in the surrounding tissue. This is particularly important in spine surgery where it is vital to protect neural structures such as the spinal cord. The limit value of radiation before permanent damage to the spinal cord is 50 GY [2,13]. In order to preserve the neural structures, a safety margin is set between the tumor tissue and the surrounding tissue. These borders are calculated with CT scans used for radiotherapy treatment planning [14]. New and more advanced radiation techniques have been developed due to improvements in image guidance such as stereotactic body radio therapy (SBRT). With SBRT, a higher dose is given in one or a few fractions. With SBRT, it becomes even more crucial to conduct precise scans for treatment planning and present compelling arguments for using CFR-PEEK over traditional titanium instrumentation [12,15].

The largest comparable clinical study was carried out by Cofano et al. This was a retrospective, single-center study of 78 patients who underwent surgery for cervical, thoracic, or lumbar metastatic lesions. Three patients were treated with cervical anterior instrumentation, which is a different instrumentation than posterior instrumentation. Hence, the actual sample size was 75. The patients were divided into two groups with 35 patients treated with carbon instrumentation and 40 with titanium instrumentation [9]. The study was well conducted but limited by the relatively small sample size. As in our study, Cofano et al.’s study showed no significant differences in terms of complications or survival between the two groups [9].

A retrospective cohort study published in 2017 by Boriani et al. included 34 patients with either thoracic or lumbar metastases or primary cancer in the spine requiring treatment with a combination of surgery and radiotherapy. Here, it was also found that thoracic/lumbar spinal fixation using CFR-PEEK implants in MSCC patients is comparable to titanium regarding stability, functionality, and the number of complications [13].

One of the major differences between the previous studies and the current cohort study is the size of the patient population (163 vs. 75 and 34, respectively). Moreover, the difference in post-surgery mean survival in the two patient groups, which was found to be non-significant in this study, was not included in the above-mentioned studies. In the study by Cofano et al., there was more detailed data collection for each patient including the grade of instability (SINS), the grade of epidural compression (ESCC score), as well as the pre- and postsurgical level of axial pain and neurological status, which should result in a more thorough patient evaluation. The results of that study were a longer mean surgical duration and a higher mean blood loss in the CI-group compared to the TI-group. The reason for this was suggested to be a more complex closing system in the carbon-based implants and the fact that more surgeries included circumferential decompression and debulking into the corpus vertebra. We found that the peri-operative blood loss in the CI-group was significantly lower than in the TI-group. The difference in the two groups amounted to a mean value of approximately 200 mL, and it could be speculated that this reduction in blood loss is beneficial in a fragile patient population. However, it is not known if the reduction in blood loss affected long-term survival.

The mechanical properties of the CFR-PEEK system were shown to be comparable to those of titanium-based systems, and the overall benefit of CFR-PEEK in orthopedic implants in terms of durability has also been strongly supported in a systematic review from 2014 by Li et al. [16].

The results from the current study underline the safety of CFR-PEEK implants and their comparability to titanium in terms of peri-operative bleeding, the duration of surgery, post-surgical complications, and average survival time.

### Strengths and Weaknesses of This Study

The primary strength of this cohort study is the relatively large patient population. Furthermore, there were no statistically significant difference between the two groups regarding gender, age, BMI, and ASA-score. This should minimize confounding and thereby make groups comparable regarding mortality, mean blood loss, the duration of surgery, and complications. Another strength of this study is that all patients were treated by a small and consistent group of surgeons at the same center. We did not specify the amount of experience of the surgeons who performed the procedures but all surgeons at the Spine Unit, Rigshospitalet, have many years of experience in complex spine surgery. The surgeons’ high level of experience is supported by the fact that the surgery time did not change during the study period, indicating that there was no learning curve associated with the introduction of carbon-based implants.

A limitation of this study is that it is not randomized, and that the surgeons’ preference decided whether patients were treated with carbon-based or titanium-based implants.

Another limitation is that only the ASA-score was used for the preoperative evaluation of the patients’ physiological status. The ASA-score is a classification system created by the American Society of Anesthesiologists. It is used to categorize the physiological status of a patient pre-surgery based on the patient’s comorbidities. Although widely used, the score has certain weaknesses such as a possible significant variation in the classification assigned to patients as well as the assumption that psychical fitness and age are unrelated [17]. A more detailed evaluation of the physiological status of the patients, taking into account factors like comorbidities, the grade of instability and epidural compression, and the pain level and neurological status of the patients, could result in a more thorough patient evaluation and hence more specific results. Another possible weakness could be that the Tokuhashi score used for estimating a patient’s survival time was not applied. Finally, the fact that the two patient groups were not matched can also be considered a limitation. Matched patient groups in terms of, e.g., age, sex, primary oncological diagnosis, and tumor level could result in a more exact comparison of the two groups and hence more representative results.

In future studies, more focus should be on the clinical impact of using CFR-PEEK implants, with regard to the benefits of better radiolucency and the radiotherapeutic effects of using CFR-PEEK implants, as also stated by Khan and Takayanagi et al. [11,12].

## 5. Conclusions

Based on a large cohort, we found that surgical treatment with CFR-PEEK for MSCC is safe and an equally sufficient treatment when compared to the more traditional titanium implants. This is in line with what previous studies have indicated. The use of CFR-PEEK could lead to improvements in the oncological treatment of patients surgically treated for MSCC.

## Figures and Tables

**Figure 1 cancers-16-00736-f001:**
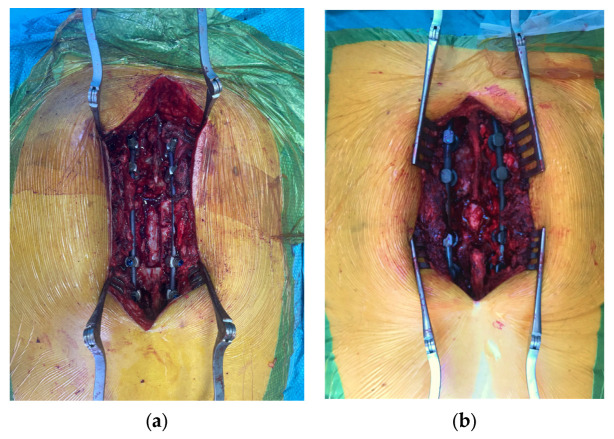
Peri-operative pictures of titanium instrumentation (**a**) and carbon instrumentation (**b**).

**Figure 2 cancers-16-00736-f002:**
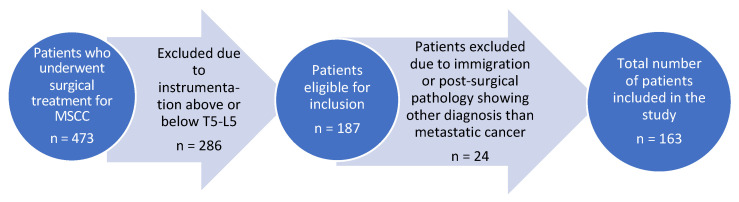
Flowchart of study population.

**Table 1 cancers-16-00736-t001:** Baseline characteristics.

	CI-Group *	TI-Group **	Total	*p*-Value
*Number of patients*	80	83	163	-
*Female/male%*	50/50	43/57	46.5/53.4	0.317
*Age (mean, years)*	66.7	66.9	≈67	0.583
*BMI (mean, kg/m^2^)*	24.8	25.4	25.1	0.478
*ASA-score (mean)*	2.7	2.7	≈2.7	0.930

* Patient group stabilized using CFR-PEEK implants; ** Patient group stabilized using titanium implants.

**Table 2 cancers-16-00736-t002:** Primary oncologic diagnosis in the carbon instrumentation (CI) and titanium instrumentation (TI) group.

Oncologic Diagnosis	CI-Group *	TI-Group **	Total
*Pulmonary cancer*	13	18	31
*Breast cancer*	14	11	25
*Renal cancer*	4	14	18
*Prostate cancer*	6	7	13
*Myeloma*	7	9	16
*Lymphoma*	12	3	15
*Colon cancer*	5	1	6
*Uterine cancer*	1	2	3
*Other ****	12	15	27
*Unknown*	6	3	9
*One tumor level and five instrumentation levels*	75	69	144
*Two tumor levels and six instrumentation levels*	5	14	19

* Patient group stabilized using CFR-PEEK implants; ** Patient group stabilized using titanium implants; *** Other = cholangiocarcinoma, c. vesicae, c. pancreatis, NEC, c. recti, c. duodeni, c. hepar, c. testis, c. oesophagus, c. ovarii, melanoma, and adrenal cancer.

**Table 3 cancers-16-00736-t003:** Causes of surgical revision in the carbon instrumentation (CI) and titanium instrumentation (TI) group.

Cause of Surgical Revision	CI-Group	TI-Group
*Implant failure*	2	1
*Post-surgical infection*	4	2
*Other indications **	0	7

* Progression of cancer, indication for vertebrectomy, epidural recompression, or wound rupture.

## Data Availability

The data of this study are available upon reasonable request from the corresponding author.

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
