# Peer review of "Carbon Instrumentation in Patients with Metastatic Spinal Cord Compression"

_cancers, 2024, doi:10.3390/cancers16040736_

Round 1

Reviewer 1 Report

Comments and Suggestions for Authors

Interesting and well structured article. Discussion is a bit short and a paragraph about the possible advantages of Ci implants should be described in more detail (advantages in radiotherapy).

Comments on the Quality of English Language

Well written article. Nothing special. Minor revision.

Author Response

Dear Reviewer

On behalf of the authors, I want to thank you for a thorough review of the manuscript. We have addressed the questions raised and hereby submit specific answers and corrections made to the manuscript. 

Discussion is a bit short and a paragraph about the possible advantages of Ci implants should be described in more detail (advantages in radiotherapy).

To expand the discussion and to elaborate on the advantages of CI implants we have expanded the discussion with the following section in line 184-197:

“Radiation sensitive metastases are treated postoperatively with radiation therapy. The goal is to target the tumor tissue with radiation and at the same time keeping the doses as low as possible in the surrounding tissue. This is particularly important in spine surgery where it is vital to protect neural structures as the spinal cord. The limit value of radiation before permanent damage to the spinal cord is 50 GY16,2. In order to preserve the neural structures, a safety margin is set between the tumor tissue and the surrounding tissue. These borders are calculated with CT scans used for radiotherapy treatment planning17. New and more advanced radiation techniques have been developed due to improvements in image guidance such as stereotactic body radio therapy (SBRT). With SBRT a higher dose is given in one or few fractions. SBRT could make even more important to be able to make precise scans for treatment planning and make stronger arguments for using CFR-PEEK instead of traditional titanium instrumentation12,18. “

Kind regards

Søren Schmidt Morgen MD, PhD

Reviewer 2 Report

Comments and Suggestions for Authors

Titanium screw and rod systems are widely used, then, why do we need to choose carbon without no specific superiority?

Comments on the Quality of English Language

Ok

Author Response

Dear Reviewer

On behalf of the authors, I want to thank you for a thorough review of the manuscript. We have addressed the questions raised and hereby submit specific answers and corrections made to the manuscript. 

Titanium screw and rod systems are widely used, then, why do we need to choose carbon without no specific superiority?

We have chosen to examine performance and safety of CI as CI is expected to decrease the deflection of radiation and improve diagnostic imaging and radiotherapy when compared to titanium implants.

Kind regards

Søren Schmidt Morgen MD, PhD

Reviewer 3 Report

Comments and Suggestions for Authors

Summary

The authors present the results of a retrospective, single-center series evaluating outcomes of 163 patients with metastatic epidural spinal cord compression undergoing spinal instrumentation with carbon-fiber implants vs. titanium implants.

Strengths

The topic is relevant to the field of spine surgery and radiation oncology.

Weaknesses/Concerns

1.       In the Introduction section, the statement, “This may result in increased risk of progression or a postponed discovery of tumor regrowth” is speculative and not supported by the citation provided. I would recommend it be clarified that there is currently no evidence to support this idea.

2.       Why were patients who had instrumentation outside of T5 – L5 excluded? This appears to be a large segment of the initial treatment population (286 patients).

3.       In the Discussion section, it is stated that, “The results of this study were a longer mean surgical duration and a higher mean blood loss in the CI-group compared to the TI-group”, but you mention in the Results section that blood loss was lower in the CI group.

Author Response

Dear Reviewer

On behalf of the authors, I want to thank you for a thorough review of the manuscript. We have addressed the questions raised and hereby submit specific answers and corrections made to the manuscript. 

The topic is relevant to the field of spine surgery and radiation oncology.

We thank the reviewer for this comment.

In the Introduction section, the statement, “This may result in increased risk of progression or a postponed discovery of tumor regrowth” is speculative and not supported by the citation provided. I would recommend it be clarified that there is currently no evidence to support this idea.

We agree that this can be considered speculative and have changed the sentence to:

It could be speculated that this could lead to postponed discovery of tumor regrowth.

Inserted in line 56-57.

Why were patients who had instrumentation outside of T5 – L5 excluded? This appears to be a large segment of the initial treatment population (286 patients).

We agree that it was not clear from the original manuscript why we chose to exclude patients with instrumentation outside of T5 – L5. We have inserted the following in the Material and Method section to explain this in line 91-96:

The CI system we use is not designed for cervical instrumentation. CI rods are rigid and come with various prebend curvatures. They cannot always be precisely adapted to the body's anatomy as it can be done with TI. This can be a particular challenge, especially in instrumentation at the pelvis or high thoracic area where there is increased kyphosis. For this reason, only patients who received instrumentation between T5-L5 were included.”

In the Discussion section, it is stated that, “The results of this study were a longer mean surgical duration and a higher mean blood loss in the CI-group compared to the TI-group”, but you mention in the Results section that blood loss was lower in the CI group.

The perioperative blood loss was lower in the CI group when compared to the TI group. We have stated that in the abstract, in the results and again in the discussion.

Kind Regards

Søren Schmidt Morgen, MD, PhD

Reviewer 4 Report

Comments and Suggestions for Authors

Dear authors, thank you for the opportunity to review your article, which I find very interesting. I would like the materials and methods section to explain better the choice of surgical treatment and whether there is a correlation with the need for radiotherapy treatment. I also think it is important to indicate in which cases a biopsy of the vertebral soma has been performed. Review the graphics in Table 2.

Author Response

Dear Reviewer

On behalf of the authors, I want to thank you for a thorough review of the manuscript. We have addressed the questions raised and hereby submit specific answers and corrections made to the manuscript. 

Dear authors, thank you for the opportunity to review your article, which I find very interesting. I would like the materials and methods section to explain better the choice of surgical treatment and whether there is a correlation with the need for radiotherapy treatment. I also think it is important to indicate in which cases a biopsy of the vertebral soma has been performed.

We agree that a more detailed description of the surgical treatment will improve the manuscript and we have elaborated on the choice of surgical treatment in the methods section line 103-106:

Patient information regarding age, gender, BMI, ASA-score, and primary oncologic diagnosis was collected at baseline and adjusted in concordance with results of the pathological perioperative tissue samples made during surgery.

It is further mentioned that we need to clarify in which cases biopsy has been taken.

In the revised manuscript we have added that all patients had perioperative biopsies taken and that these were analyzed at the pathological department:

In all cases tissue samples were sent for analysis at the pathological department and primary cancer diagnose were registered if not known beforehand.

Inserted in line 113-115.

Review the graphics in Table 2.

We have reviewed the graphics in table 2. We have added the levels of instrumentation and the number of metastatic levels in each patient.

We have added a section to the discussion after the update of table 2. Line 166-170:

“There were more patients in the TI group who had two levels of metastases. These patients underwent instrumentation at six levels instead of five levels. This could explain some of the difference in blood loss between the two groups, since longer constructs and longer wounds potentially lead to a higher blood loss.”

Kind regards

Søren Schmidt Morgen, MD, PhD

Round 2

Reviewer 2 Report

Comments and Suggestions for Authors

I personally think that at least the authors should add a representative demonstration of surgical procedure as one Figure. This would surely enhance the readability and understanding of the study.

Author Response

The two surgical procedures are illustrated in Figure 1a and Figure 1b.

Figure 1a and 1b. Illustrates perioperative pictures of titanium instrumentation 1a and carbon instrumentation 1b.